# A unified model of human hemoglobin switching through single-cell genome editing

Yong Shen[1,2,3], Jeffrey M. Verboon[1,2,3], Yuannyu Zhang[4], Nan Liu [1,2], Yoon Jung Kim[4], Samantha Marglous[1,2,3,5], Satish K. Nandakumar [1,2,3], Richard A. Voit[1,2,3], Claudia Fiorini [1,2,3], Ayesha Ejaz[1,2,3], Anindita Basak [1,2,3], Stuart H. Orkin[1,2,5,6], Jian Xu [4] & Vijay G. Sankaran [1,2,3,5✉]

Key mechanisms of fetal hemoglobin (HbF) regulation and switching have been elucidated through studies of human genetic variation, including mutations in the *HBG1/2* promoters, deletions in the β-globin locus, and variation impacting BCL11A. While this has led to substantial insights, there has not been a unified understanding of how these distinct genetically-nominated elements, as well as other key transcription factors such as ZBTB7A, collectively interact to regulate HbF. A key limitation has been the inability to model specific genetic changes in primary isogenic human hematopoietic cells to uncover how each of these act individually and in aggregate. Here, we describe a single-cell genome editing functional assay that enables specific mutations to be recapitulated individually and in combination, providing insights into how multiple mutation-harboring functional elements collectively contribute to HbF expression. In conjunction with quantitative modeling and chromatin capture analyses, we illustrate how these genetic findings enable a comprehensive understanding of how distinct regulatory mechanisms can synergistically modulate HbF expression.

[1] Division of Hematology/Oncology, Boston Children's Hospital, Harvard Medical School, Boston, MA, USA. [2] Department of Pediatric Oncology, Dana-Farber Cancer Institute, Harvard Medical School, Boston, MA, USA. [3] Broad Institute of MIT and Harvard, Cambridge, MA, USA. [4] Children's Medical Center Research Institute, Department of Pediatrics, Harold C. Simmons Comprehensive Cancer Center, University of Texas Southwestern Medical Center, Dallas, TX, USA. [5] Harvard Stem Cell Institute, Cambridge, MA, USA. [6] Howard Hughes Medical Institute, Boston, MA, USA. ✉email: sankaran@broadinstitute.org

The regulation of fetal hemoglobin (HbF) has been of substantial interest, both for its value to enable improved therapies to elevate HbF as a treatment in sickle cell disease and β-thalassemia, as well as for its broader implications as a paradigm for understanding the developmental control of gene expression[1–3]. A number of studies have provided insights into HbF regulation through the identification and analysis of naturally occurring mutations impacting this process. Such variants have been extensively characterized at two distinct loci: (1) in the gene encoding the BCL11A transcription factor and (2) within the β-globin gene locus that harbors the HbF genes, *HBG1* and *HBG2*. Both common and rare variants in the *BCL11A* gene alter HbF expression in erythroid cells, with rare loss-of-function variants resulting in substantially increased HbF[4–10]. Other studies focused on the β-globin locus have identified a number of single-nucleotide variants (SNVs) and small deletions in the

*HBG1* and *HBG2* proximal promoters that allow upregulation of HbF levels to varying extents (Fig. 1a and Supplementary Data 1)[11,12]. Recent studies have begun to elucidate how specific variants in these proximal promoters act by either preventing or facilitating the interactions of *trans*-acting regulatory factors, most notably the key HbF silencing factors BCL11A and ZBTB7A, with specific sequences[13–16]. In addition to variants affecting the *HBG1/2* proximal promoters, large deletions that span the entirety of the adult β-globin *HBB* and *HBD* genes also increase HbF expression to varying extents. Such deletions can be broadly classified into two categories: those that have higher *HBG1/2* mRNA and therefore HbF production, termed hereditary persistence of fetal hemoglobin (HPFH) deletions, and those that are characterized by lower HbF production with resultant globin chain imbalance, termed δβ-thalassemia (Fig. 1a and Supplementary Fig. 1a). We and others have suggested that a 3.5 kb

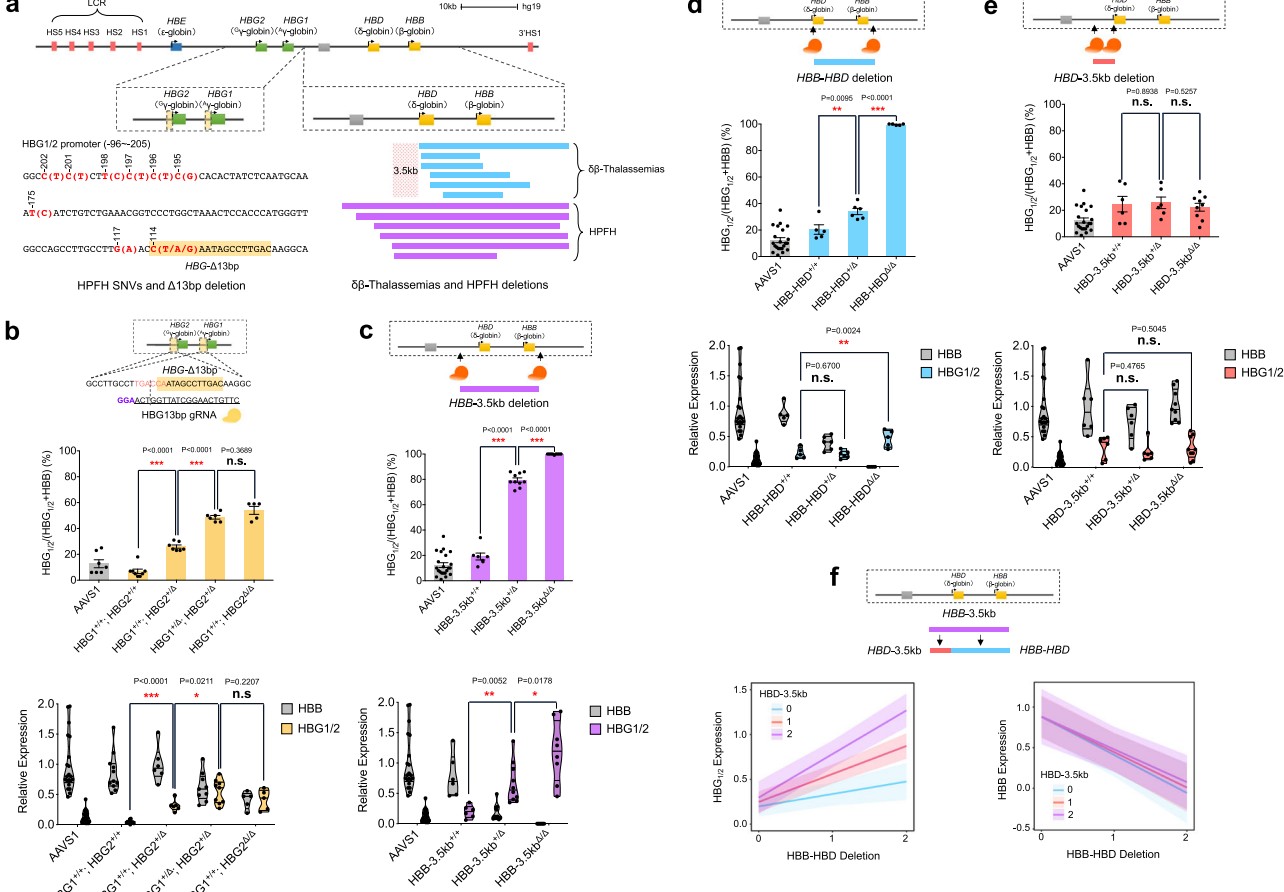

**Fig. 1 Modulation of HbF regulation through recapitulation of specific *cis*-regulatory element perturbations in single cells. a** Schematic of HPFH SNVs and *HBG*-Δ13bp deletion at the proximal γ-globin promoter and large deletions within the β-globin locus that have been reported in individuals with δβ-thalassemias (blue) and HPFH (purple). SNV single-nucleotide variant, HPFH hereditary persistence of fetal hemoglobin, *HBG*-Δ13bp 13 bp deletion at the proximal *HBG1/2* promoter −101 to −114. **b** Gene expression analysis for γ-globin (*HBG1/2*) and β-globin (*HBB*) mRNA in erythroid burst-forming units (BFU-E) derived from HSPC-derived erythroblasts upon genome editing of *HBG*-Δ13bp region in the *HBG1*, *HBG2*, or both promoters, n = 3 biologically independent experiments. Results are shown as mean ± SEM (P values are labeled on the top of each comparison. *P < 0.05, ***P < 0.001, n.s. statistically non-significant by a two-tailed Student's t test). HBG1Δ, HBG2Δ: editing mutation within *HBG*-Δ13bp region. **c–e** Globin gene expression analysis in BFU-E upon genetic perturbations of elements composing the entire *HBB*-3.5kb deletion, n = 6 biologically independent experiments (**c**); *HBB*-*HBD* deletion, n = 3 biologically independent experiments (**d**); an *HBD*-3.5kb deletion, n = 3 biologically independent experiments (**e**). *HBB*-3.5kbΔ: region deletion starting from HBD upstream 3.5 kb to HBB 3' end; *HBB*-*HBD*Δ: region deletion starting from *HBD* TSS to *HBB* 3' end; *HBD*-3.5kb: deletion starting from *HBD* upstream 3.5 kb to *HBD* TSS. Results are shown as mean ± SEM (P values are labeled on the top of each comparison. *P < 0.05, **P < 0.01, ***P < 0.001, n.s. statistically non-significant by a two-tailed Student's t test). **f** Quantitative modeling on *HBG1/2* and *HBB* mRNA expression from genetic perturbations of elements composing *HBB*-*HBD* deletion combined with (red line and purple line)/without (blue line) *HBD*-3.5kb deletion using a linear mixed model. *HBB*-*HBD* deletion (0: +/+; 1: +/Δ; 2: Δ/Δ); *HBD*-3.5kb deletion (0: +/+; 1: +/Δ; 2: Δ/Δ). Modeling lines and bands are shown as mean ± 95% CI.

region upstream of the *HBD* gene may underlie the difference between these two groups of deletions, although this remains to be functionally tested[17–19].

Despite the substantial knowledge that has arisen through these studies, which have primarily focused on one type of genetic variant or another in isolation, a holistic view of HbF regulation has yet to emerge. Recent work has cataloged and defined biochemical interactions and epigenetic marks that occur across the β-globin locus, including several binding sites for BCL11A and binding sites within the *HBG1/2* promoters for ZBTB7A (Supplementary Fig. 1b). However, a unified model integrating how these various regions of the β-globin locus interact has not yet been produced. One impediment to achieving this goal has been the inability to study humans with combinations of these rare variants that have a major impact on HbF levels. Furthermore, a number of experimental limitations have constrained potential insights in cellular models. While genome editing in transformed erythroid cell lines has enabled clonal analysis, some observations made in primary human hematopoietic cells cannot be faithfully recapitulated in this context[18]. Moreover, despite the fact that genome editing in primary hematopoietic stem and progenitor cells (HSPCs) has progressed significantly[20], such perturbations create a heterogeneous array of edits that can typically only be analyzed in bulk.

Here we sought to address these limitations in order to capture a more unified view of HbF regulation. We began by developing a system using genome editing capable of recapitulating specific mutations either individually or in combination. By integrating this with functional analysis of distinct genome edits in the progeny of single human HSPCs, we are able to assess the specific outcome of these edits upon HbF regulation at single-variant resolution. Through quantitative modeling of this data, we illuminate functional genetic interactions between specific perturbations involving these regulatory elements. We bolster these findings through the biochemical analysis of locus-specific long-range chromatin interactions using the CRISPR affinity purification in situ of regulatory elements (CAPTURE) approach and with chromosome conformation capture (3C) in the presence or absence of BCL11A[21,22]. These holistic analyses provide previously unappreciated insights into the overarching mechanisms necessary for HbF expression and switching. More broadly, we also illuminate the value of performing single-cell individual and combinatorial functional perturbations that are inspired by human genetic variation.

## Results

**A single-cell genome-editing functional assay for globin gene regulation.** We initially aimed to create a system to address two major limitations that exist in this field. First, while transformed clonal erythroid cell lines enable analysis of specific regulatory elements, perturbations in these cell models may not always faithfully recapitulate in vivo observations[18]. Second, combinations of these rare genetic perturbations are almost never observed in humans, thus limiting inferences that can be made using in vivo data[11]. To overcome these limitations, we sought to create specific genetic perturbations guided by natural genetic variation through genome editing in an isogenic setting using human HSPCs from healthy donors. These could then be plated at low density in semi-solid medium to form erythroid colonies (burst-forming unit erythroid (BFU-E) colonies) from which cells derived from a single progenitor could be isolated for paired RNA and DNA analyses[23]. We have previously shown that single-cell analysis of individual cells derived from BFU-E colonies demonstrate close clonal relationships, as assessed by sharing of mitochondrial DNA mutations, and each of the single

erythroblasts from any BFU-E cluster closely together using gene expression profiling, irrespective of donor[24]. Therefore, by isolating the hundreds to thousands of differentiated erythroblasts in each BFU-E colony derived from a single distinct edited HSPC, we could obtain information on the specific genotype present in that colony and simultaneously interrogate the impact on globin gene regulation in the same colony (Supplementary Fig. 2). We hypothesized that this approach would enable us to study the impact of each individual genetic mutation and combinations of these perturbations to gain a more complete understanding of HbF regulation. Furthermore, we expected that existing genome-editing tools would enable us to recreate each category of the well-characterized rare perturbations that significantly alter HbF, including variants in the proximal promoters of the *HBG1/2* genes, deletions involving *HBD* and *HBB*, and perturbation of the key *trans*-acting regulatory factors, BCL11A and ZBTB7A (Fig. 1a and Supplementary Fig. 1b).

**Modulation of HbF regulation by recapitulating specific *cis*-regulatory element perturbations in the progeny of single cells.** Given the recent identification of key sequence elements that are bound by BCL11A in the *HBG1* and *HBG2* promoters[15,16,25,26], we first sought to test this functional assay by recapitulating the 13 bp deletion of these elements that has been observed in vivo[27]. In bulk, we observed efficient editing and a high frequency of the 13 bp deletion through sequencing analysis (Supplementary Fig. 3a–c). There was robust induction of *HBG1/2* without notable perturbation of erythroid differentiation or maturation (Supplementary Fig. 4a, b, d). By screening hundreds of colonies, we could obtain numerous independent clones targeted by either the *HBG1/2* promoter guide or with a control *AAVS1*-targeting guide, which is a genomic region that has no impact on hematopoiesis upon editing[28]. We initially noted that in *AAVS1*-edited colonies, there was minimal donor-to-donor variability observed in globin mRNA levels enabling pooling of data across multiple independent donors (Supplementary Fig. 5). By either separating the colonies that harbored disruptive deletions in either one or both *HBG1* or *HBG2* promoters, as well as through aggregate analysis across these homologous genes, we found that each deletion derepressed *HBG1/2* mRNA expression in an additive manner, without signs of resultant globin chain imbalance (Fig. 1b and Supplementary Figs. 6a and 12). Given similar impacts seen with editing at either the *HBG1* or *HBG2* promoters, we performed aggregate analysis with these edits together (Supplementary Fig. 6a) and treat these similarly in subsequent modeling. Importantly, we ensured that we did not analyze deletions or inversions created across the *HBG1/2* region through targeted sequencing approaches. Our findings provide functional validation for the recently emerging models that BCL11A acts through a sequence element that overlaps the 13 bp deletion to silence the *HBG1/2* genes, while competing with activating factors such as NF-Y, and that each element in the proximal promoters of these homologous genes appears to act in a semi-autonomous manner[15,16,25].

We next sought to use the single-cell functional assay to dissect how larger HPFH or δβ-thalassemia deletions may impact HbF silencing. We initially targeted the region from 3.5 kb upstream of *HBD* that is commonly removed in many HPFH deletions and concomitantly removed the region spanning the adult *HBD* and *HBB* genes, which is deleted in nearly all HPFH and δβ-thalassemia deletions (Fig. 1b and Supplementary Figs. 3a, d, e and 4a, b, e). This resulted in robust *HBG1/2* expression in both heterozygous and homozygous states without any signs of globin chain imbalance. We additionally examined clones with an inversion of this element, which displayed globin chain imbalance

in contrast to the deletion, despite some elevation of *HBG1/2* expression (Supplementary Fig. 6b). To gain further insights into the elements that comprise this larger region and how they may function independently, we separately removed the region spanning the adult *HBD* and *HBB* genes, which as expected led to a phenotype characterized by imbalance of the beta-like to alpha-like mRNAs with minimal HbF induction in the heterozygous state (Fig. 1d), or the *HBD* upstream 3.5 kb region—which is by itself never removed in any in vivo deletions causing HPFH —that resulted in little HbF induction (Fig. 1e). This data suggested that there is an interaction between these two deleted regions such that the increase in HbF when both regions are deleted is much greater than would be expected by either deletion alone. These findings provide support for the promoter competition model that has been proposed to underlie hemoglobin switching, whereby interaction between the locus control region (LCR) enhancer with the adult globin promoters competes with LCR interactions with the *HBG1/2* promoters[3].

To quantitatively test this, we fit a linear mixed model (LMM) for *HBG1/2* and *HBB* mRNA expression to a dosage (0, 1, 2) for each partial deletion, wherein the larger deletion would consist of both. We also treated the interaction term between the deletions as a fixed effect in the LMM. Moreover, we allowed for random intercepts to account for donor and guide RNA (gRNA)-based variation when applicable. Importantly, when looking at the model fit to *HBG1/2* mRNA expression, we observe significant reinforcement interactions (synergy) between these regions in the model ($\beta = 0.17$; *P* value < 0.0001), which can be noted visually by the non-parallel lines of the interaction plot with increasing slopes as dosages of each deletion increase (Fig. 1f). Conversely, when modeling *HBB* expression, we see that only the *HBD-HBB* deletion impacts *HBB* mRNA expression and there is no effect on this by the *HBD*-3.5 kb region ($\beta = 0.03$; *P* value = 0.55) (Fig. 1f). Collectively, this analysis revealed key *cis*-regulatory elements in both the *HBG1/2* proximal promoters and in long-range distal elements near the adult globin genes that are critical for HbF silencing. However, we wanted to gain further insights into how these elements could interact with the key and genetically nominated *trans*-acting regulators of HbF, BCL11A, and ZBTB7A.

**Insights from perturbation of BCL11A and ZBTB7A in single progenitor-derived erythroid cells**. We therefore next sought to characterize the impact of perturbing BCL11A alone in our single cell functional assay. Rare haploinsufficient mutations in BCL11A have been shown to enable significant persistence of HbF[8–10]. To mimic the impact of these rare variants, we screened several gRNAs and identified two that showed efficient cutting as assessed by indel frequency (62 and 81% editing efficiency for guides targeting exon 2 and 4, respectively) (Supplementary Fig. 7a). In bulk, we found no significant perturbation of erythroid maturation and differentiation, while observing robust HbF induction with both of the guides (Supplementary Fig. 7e). Interestingly, perturbation of exon 4 appeared to result in much higher HbF induction than would be expected for the degree of editing achieved compared to exon 2 (Supplementary Fig. 7b–d). This prompted us to examine the observed protein expression levels. While targeting of exon 2 resulted in a reduction of overall BCL11A levels, targeting of exon 4 resulted in an increase in the levels of a truncated protein form of BCL11A, as would be predicted from the commonly observed indels with this gRNA (Supplementary Fig. 7a, b). These results suggest that targeting of exon 4 creates frameshift mutations that bypass nonsense mediated decay pathways, given its location near the C-terminal end of the protein, and thereby result in the production of dominantly interfering forms of BCL11A that may inhibit activity of a portion

of the remaining wild-type (WT) allele of *BCL11A*. To directly demonstrate dominant negative activity, we expressed the cDNAs encoded by these frameshift mutations and observed increased *HBG1/2* mRNA production in adult erythroid cells, which is distinct from the *HBG1/2* repression observed with over-expression of the WT cDNA (Supplementary Fig. 8). This finding suggests a possible role for BCL11A homodimerization in HbF regulation[29,30]. Alternatively, this dominantly interfering form may sequester co-repressors, given that DNA binding by BCL11A requires zinc fingers 4-6, which would be removed by these deletions[15,31].

We transitioned to single cell functional analyses of these edits to assess the consequences across a range of perturbations impacting BCL11A. We observed robust induction of HbF in cells with heterozygous edits in single HSPCs that underwent erythroid differentiation. We observed an increase in these levels from a baseline of ~20% to ~60% *HBG1/2* with heterozygous edits, and observed ~94% with homozygous edits at this exon (Supplementary Fig. 9a). Interestingly, with perturbation of exon 4, we observed ~80% *HBG1/2* with heterozygous edits and ~96% with homozygous edits in single progenitors (Supplementary Fig. 9b). These observations bolstered the data obtained in bulk and enabled us to illuminate the critical role of BCL11A in HbF silencing in primary cells at single-cell resolution, while also establishing an allelic series for BCL11A perturbations. The exon 4 edits enabled us to alter activity to a level lower than haploinsufficiency, given dominant negative activity, which together with exon 2 edits provided an allelic series in *BCL11A* to use in interrogating functional interactions. We do note that all loss-of-function variants that have been identified in patients with *BCL11A* haploinsufficiency to date occur in more N-terminal regions of the protein in comparison to the BCL11A exon 4 guide we selected[10], consistent with the resultant dominant interference by indels in this region.

While mutations in ZBTB7A have not been found in individuals with persistence of HbF, this *trans*-acting factor has been genetically-nominated through analysis of *cis*-regulatory elements to which it binds that are disrupted by mutations associated with elevated HbF levels, including the $-195C > G$ mutation[25]. As a result, we also disrupted *ZBTB7A* with gRNAs and examined the consequences of this perturbation in our single cell functional assay (Supplementary Fig. 10). Consistent with the restricted terminal erythroid defects observed with ZBTB7A perturbation[14,32], we found ostensibly normal erythroid maturation in the targeted cells (Supplementary Fig. 10e, f). With this perturbation, we noted an elevation of *HBG1/2* as a percentage of total β-like globin mRNAs. Interestingly, there was significant derepression of *HBG1/2*, beyond what is observed from reciprocal *HBB* silencing (Supplementary Fig. 11). This observation suggests that ZBTB7A has a distinct role in silencing *HBG1/2* and maintaining appropriate expression levels. The significant and out of proportion enhancement of *HBG1/2* mRNA expression compared to the decrease in *HBB* mRNA upon ZBTB7A perturbation suggests a key role for ZBTB7A in autonomous silencing of the *HBG1/2* genes.

**Functional genetic interactions between BCL11A, ZBTB7A, and β-globin locus *cis*-regulatory elements**. Given the findings from our single cell functional analysis of individual perturbations, we next sought to combine these perturbations to assess for functional genetic interactions between these distinct regulatory factors. We initially began by combining the perturbation of the *HBG1/2* promoter 13 bp element (herein, Δ13 bp) with targeting of BCL11A, which has been shown to bind to this region[15,25]. Interestingly, combined perturbation of Δ13 bp and BCL11A

showed non-additive effects across a range of distinct genotypes, suggestive of a functional interaction (Fig. 2a, c and Supplementary Fig. 12). We applied a LMM to directly quantify this interaction by modeling the impact of individual or combined perturbations upon *HBG1/2* and *HBB* mRNA expression. We treated the various *BCL11A* perturbations as an allelic series based upon their effects with heterozygous exon 2 edits given a value of 1, heterozygous exon 4 edits given a value of 2 (given greater than haploinsufficient perturbation), and homozygous edits of BCL11A given a value of 3 ("Methods"). As our previous results had suggested a purely additive effect for *HBG1/2* edits, these were given values of 0-4 to represent the number of *HBG1/2* proximal promoter edits observed in any single cell. Both allelic series were treated as fixed effects and fit to values of *HBG1/2* and *HBB* mRNA expression. Random intercepts were allowed, as previously described, to account for donor and other experimental sources of variation when applicable. The combined effects of BCL11A and Δ13 bp perturbations showed a clear and significant antagonistic interaction with this LMM ($\beta = -0.12$, $p$-value $< 0.0001$) (Fig. 2c). This finding shows that the *HBG1/2* promoters and BCL11A act via an overlapping and interacting pathway, thus causing non-additive effects upon perturbation, as seen by the decreasing effect (slope) of *HBG1/2* proximal

promoter edits on *HBG1/2* mRNA expression, with an increasing effect (slope) on *HBB* mRNA expression as the BCL11A allelic series is increased. As expected, when modeling *HBB* mRNA expression versus these same fixed and random effects, we find a similar, albeit moderated, result consistent with the total levels of β-globin chains staying roughly in balance (Fig. 2c and Supplementary Fig. 12). This finding provides functional validation for the previously characterized binding interaction of BCL11A with these elements (Supplementary Fig. 1b)[15,25].

We then sought to combine perturbation of the *HBB*-3.5kb element, which is commonly removed in HPFH deletions, with BCL11A. By combining these edits in single hematopoietic progenitors and deriving erythroblasts in colony assays, we again observed a non-additive effect, suggestive of a functional interaction across a range of genotypes (Fig. 2d, e and Supplementary Fig. 12). We again utilized LMMs to demonstrate a similar antagonistic interaction effect ($\beta = -0.15$; $P$ value $< 0.001$) between the *HBB*-3.5 kb deletion and BCL11A perturbation (across the full allelic series involving perturbation of exons 2 or 4) upon *HBG1/2* and *HBB* mRNA expression. This finding could readily be visualized by the decreasing effect (slope) of the *HBB*-3.5 kb edits on both *HBG1/2* and *HBB* mRNA expression as the BCL11A allelic series increases (Fig. 2f). While these elements have been hypothesized to

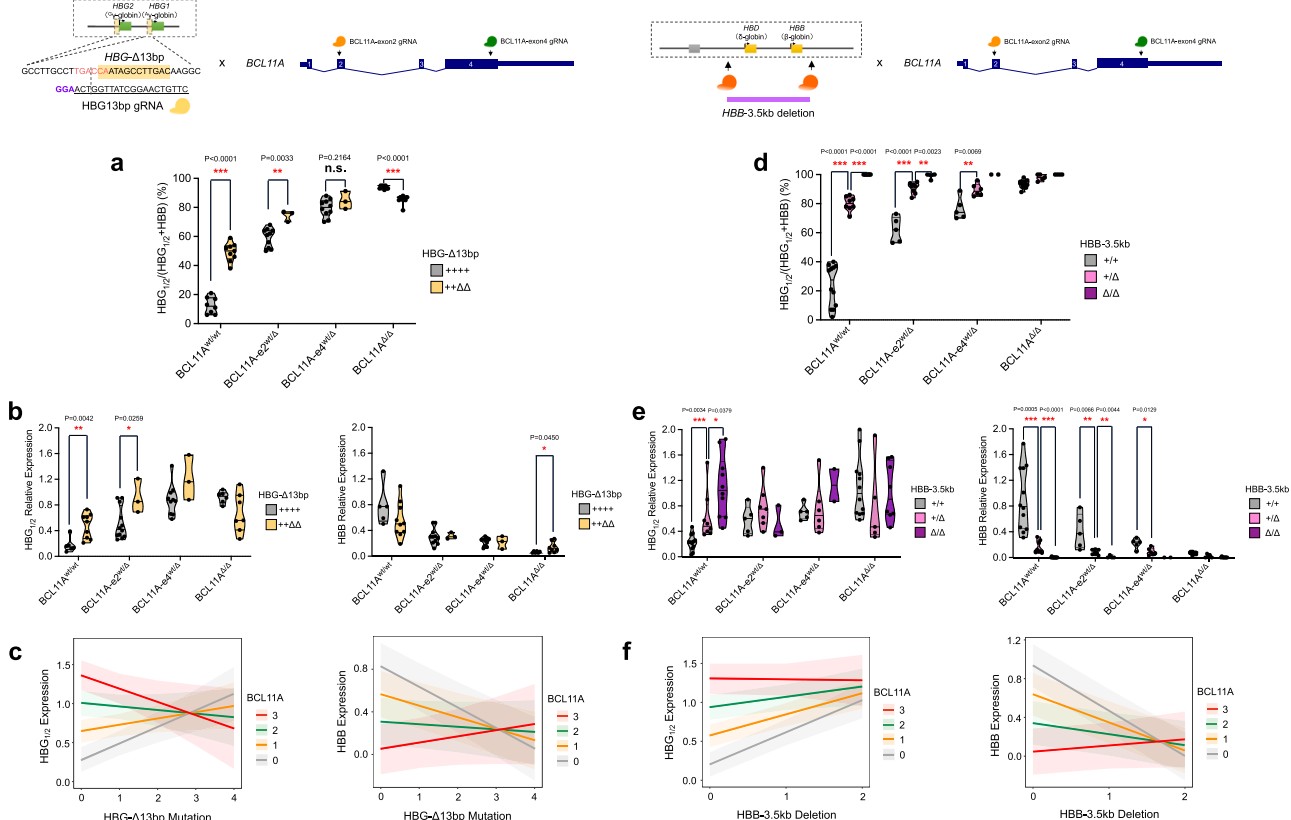

**Fig. 2 Functional genetic interactions between BCL11A and β-globin locus *cis*-regulatory elements. a, b** Globin gene expression analysis in BFU-E colony upon combined perturbation of *HBG*-Δ13bp and BCL11A. BCL11A-e2Δ: frameshift mutation in BCL11A exon 2; BCL11A-e4Δ: frameshift mutation in BCL11A exon 4; *HBG*Δ: editing mutation within *HBG*-Δ13bp region. Results are shown as violin plot (*P* values are labeled on the top of each comparison. \**P* < 0.05, \*\**P* < 0.01, \*\*\**P* < 0.001 by a two-tailed Student's *t* test). **c** Quantitative modeling on *HBG1/2* or *HBB* mRNA expression from combined perturbation of *HBG*-Δ13bp and BCL11A. *HBG*-Δ13bp mutation (0: ++++; 1: +++Δ; 2: ++ΔΔ; 3: +ΔΔΔ; 4: ΔΔΔΔ); BCL11A mutation (0: BCL11A^wt/wt^; 1: BCL11A-e2^wt/Δ^; 2: BCL11A-e4^wt/Δ^; 3: BCL11A^Δ/Δ^). Modeling lines and bands are shown as mean ± 95% CI. **d, e** Globin gene expression analysis in BFU-E colony upon combined perturbation of *HBB*-3.5kb element and BCL11A. Results are shown as violin plot (*P* values are labeled on the top of each comparison. \**P* < 0.05, \*\**P* < 0.01, \*\*\**P* < 0.001 by a two-tailed Student's *t* test). **f** Quantitative modeling on *HBG1/2* or *HBB* mRNA expression from combined perturbation of *HBB*-3.5kb element and BCL11A. *HBB*-3.5kb deletion (0: +/+; 1: +/Δ; 2: Δ/Δ); BCL11A mutation (0: BCL11A^wt/wt^; 1: BCL11A-e2^wt/Δ^; 2: BCL11A-e4^wt/Δ^; 3: BCL11A^Δ/Δ^). Modeling lines and bands are shown as mean ± 95% CI.

be critical for BCL11A-mediated silencing[17], we now directly and quantitatively demonstrate at a functional level that BCL11A and this long-range regulatory element must intersect.

We next combined perturbation of the *HBB*-3.5kb element with ZBTB7A, the latter of which is primarily suggested to act in a local autonomous manner to mediate *HBG1/2* silencing[25]. Interestingly, we observed significant attenuation of the *HBG1/2* induction caused by ZBTB7A perturbation upon concomitant perturbation of the *HBB*-3.5kb element (Fig. 3a–c), which could be quantified using the LMM ($\beta = -0.75410$; P value <0.0001). This finding demonstrates that ZBTB7A also acts through both local and long-range interactions to silence HbF. This prompted us to examine whether BCL11A and ZBTB7A were acting independently using our single-cell functional assay. Indeed, with combined perturbations of both *trans*-acting factors, we observed independent induction of *HBG1/2* expression, which could be demonstrated with the LMM and demonstration of completely parallel lines in the visualization of this model (Fig. 3d–f; $\beta = 0.16148$; P value = 0.0623).

Collectively, our findings therefore suggest that BCL11A has a critical functional role in enabling silencing via two distinct genetically nominated *cis*-regulatory elements: the local *HBG1/2* promoters and distal regulatory elements upstream of the *HBD* gene. ZBTB7A also appears to have roles in both local silencing of

the *HBG1/2* genes and in interaction with long-range regulatory elements. To gain further insights, we would ideally combine perturbations of these two distinct *cis*-regulatory elements at the β-globin locus to also assess functional interactions. However, we found that tandem perturbations involving simultaneous introduction of 3–4 gRNAs primarily resulted in larger deletions of the entire *HBG1/2* to *HBB* region, preventing this analysis.

**Analysis of distinct β-globin locus *cis*-regulatory elements and BCL11A dependence using CAPTURE and 3C assays.** We next sought to integrate the results obtained from our single-cell functional assay of globin gene regulation with insights from systematic biochemical assessments of long-range DNA interactions observed at critical mutation-associated *cis*-regulatory elements within the β-globin locus. Long-range chromatin interactions including enhancer–promoter looping play causative roles in transcriptional activation, as demonstrated by induced loop formation at the β-globin locus[33,34]. The biotinylated dCas9-based CAPTURE method enables the dissection of long-range DNA interactions at native chromatin that play critical roles in transcriptional regulation[21,22]. Given the need for significant cell numbers to achieve high-resolution multiplexed CAPTURE analyses, we initially used HUDEP-2 cells that harbor an adult-

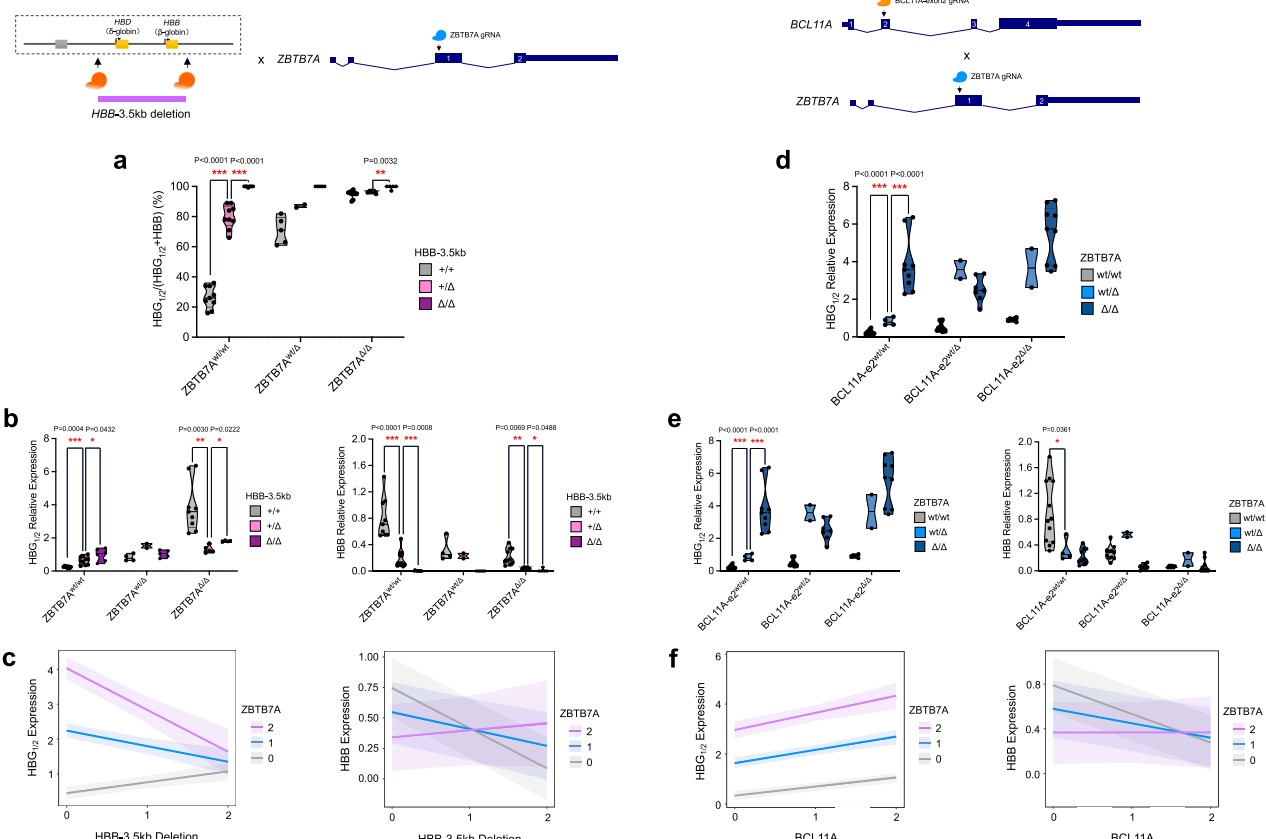

**Fig. 3 Functional genetic interactions of ZBTB7A with β-globin locus *cis*-regulatory element and BCL11A. a, b** Globin gene expression analysis in BFU-E colony upon combined perturbation of 3.5kb-*HBB* element and ZBTB7A. Results are shown as violin plot (P values are labeled on the top of each comparison. *P < 0.05, **P < 0.01, ***P < 0.001 by a two-tailed Student's t test). **c** Quantitative modeling on *HBG1/2* or *HBB* mRNA expression from combined perturbation of *HBB*-3.5kb element and ZBTB7A. *HBB*-3.5kb deletion (0: +/+; 1: +/Δ; 2: Δ/Δ); ZBTB7A mutation (0: ZBTB7A^wt/wt; 1: ZBTB7A^wt/Δ; 2: ZBTB7A^Δ/Δ). Modeling lines and bands are shown as mean ± 95% CI. **d, e** Globin gene expression analysis in BFU-E colony upon combined perturbation of BCL11A and ZBTB7A. BCL11A-e2Δ: frameshift mutation in BCL11A exon 2; ZBTB7A^Δ: frameshift mutation in ZBTB7A. Results are shown as violin plot (P values are labeled on the top of each comparison. *P < 0.05, ***P < 0.001 by a two-tailed Student's t test). **f** Quantitative modeling on *HBG1/2* or *HBB* mRNA expression from combined perturbation of BCL11A and ZBTB7A. BCL11A mutation (0: BCL11A^wt/wt; 1: BCL11A-e2^wt/Δ; 2: BCL11A-e2^Δ/Δ); ZBTB7A mutation (0: ZBTB7A^wt/wt; 1: ZBTB7A^wt/Δ; 2: ZBTB7A^Δ/Δ). Modeling lines and bands are shown as mean ± 95% CI.

like globin expression pattern and targeted knockout of BCL11A in these experiments that results in predominant expression of *HBG1/2*[15] (Supplementary Fig. 13). Using this approach, we engineered WT or BCL11A knockout HUDEP-2 cells co-expressing C-terminally biotinylated dCas9 and multiplexed single gRNAs (sgRNAs) targeting the *HBB*, *HBG1/2*, *HBD*-3.5kb, and upstream enhancer LCR (harboring hypersensitive sites (HS), 1–5) (Fig. 4). As expected, given the greatly reduced *HBB* mRNA expression upon BCL11A knockout, we observed decreased long-range DNA interactions between the LCR HS1–5 and *HBB* promoter upon this knockout with concomitantly increased interactions between the LCR and *HBG1/2* promoters (Fig. 4a, b). Remarkably, strong interactions observed in the adult globin expressing state between the silenced *HBG1/2* genes and the *HBD*-3.5kb region were completely abrogated upon deletion of BCL11A (Fig. 5a, b). These findings are concordant with the functional interaction observed between BCL11A and the *HBD*-3.5kb upstream element in HbF silencing from our single-cell analysis. These findings were further corroborated by using the *HBD*-3.5kb region as the capture bait region (Fig. 4c, d). We observed markedly reduced chromatin interactions between the *HBD*-3.5kb region and the *HBG1/2* promoters, illustrating loss or destabilized long-range chromatin interactions between *HBD*-3.5kb and *HBG1/2* in the absence of BCL11A. Of note, *HBD*-3.5kb region had significantly more long-range chromatin interactions than other *cis*-regulatory elements within the β-globin locus in WT HUDEP-2 cells, consistent with previous reports[21,22]. Importantly, the majority of these *HBD*-3.5kb region-mediated long-range interactions were lost or significantly weakened in BCL11A knockout cells, suggesting that BCL11A is required for the maintenance of proper three-dimensional (3D) organization of the β-globin locus to facilitate *HBB* transcription.

Interestingly, increased interactions of both the *HBG1/2* regions and the *HBD*-3.5kb upstream element with the 3' HS1 were noted upon knockout of BCL11A (Fig. 4a–d). This suggests that there is global reorganization and long-range alterations in chromatin conformation seen upon knockout of BCL11A. In addition, by using the LCR HS1–5 as the capture baits, we found markedly decreased interactions between the LCR and *HBD*-3.5kb region in BCL11A knockout cells (Fig. 4e, f), consistent with the results obtained using the reciprocal *HBD*-3.5kb region as the capture bait (Fig. 4c, d). These results suggest that the 3D configuration mediated by long-range chromatin interactions between LCR and *HBD*-3.5kb elements were impaired upon loss of BCL11A. Importantly, we noted decreased interactions between the LCR and the *HBB* gene and increased interactions between the LCR and the *HBG1/2* genes (Fig. 4e, f), consistent with shifts in globin mRNA expression patterns observed upon BCL11A knockout (Supplementary Fig. 13c, d). To ensure that these interaction losses were also observed in primary erythroid cells upon perturbation of BCL11A, we performed genome editing of BCL11A followed by 3C assays in bulk human HSPC-derived erythroblasts, which revealed loss of the key interactions between the *HBD*-3.5kb region and the *HBG1/2* promoters with concomitantly increased interaction between the LCR and *HBG1/2* promoters upon perturbation of BCL11A (Fig. 5a–c).

The insights from the biochemical interaction data obtained through CAPTURE and 3C assays complement the data obtained through our single-cell functional assessments, enabling the derivation of a unified model of HbF regulation based on human genetic observations (Fig. 5d, e). Specifically, our findings suggest at both a functional level and through complementary biochemical assessments that two distinct genetically nominated *cis*-regulatory regions, the *HBG1/2* proximal promoters and the *HBD* 3.5 kb upstream region, physically interact at native chromatin through long-range DNA interactions and act collaboratively

with BCL11A to silence HbF expression. Upon targeted perturbation of either of these elements or BCL11A, we see derepression of HbF silencing that is accompanied by dissociation of the locus-regulating chromatin interactions mediated by these elements. Importantly, in this unified model, we distinguish the long-range interactions mediated by BCL11A and ZBTB7A, as well as the associated *cis*-elements with the local and independent silencing of the *HBG1/2* genes by these factors. Hence, our findings establish evidence for the model that different *HBG1/2*-regulating *cis*-regulatory elements physically and functionally interact in 3D space to coordinate chromatin structure and gene transcription, providing a clearer picture of how distinct elements nominated through human genetic studies of HbF variation may function cooperatively in native biological contexts.

## Discussion

Here we have developed and utilized a single-cell functional assay with targeted genome editing in primary human HSPCs to mimic perturbations that are observed in vivo in humans. We sought to use these targeted perturbations to gain a more complete picture of the regulation of HbF, a key therapeutic target for the major disorders of β-hemoglobin, sickle cell disease and β-thalassemia[2,35]. While targeted perturbations of the 13 bp element in the *HBG1/2* promoters, the 3.5 kb region upstream of *HBD*, BCL11A, and ZBTB7A have been individually studied, how all of these distinct genetically-nominated perturbations may function together has remained unknown. Through the use of individual and combined perturbations, along with quantitative modeling of this data, we have now refined our understanding of this process. We are able to show that BCL11A has two major functions to enable effective HbF silencing: acting "locally" and through "distal" interactions. We show that the local function of BCL11A enables silencing of the *HBG1/2* genes via promoter interactions involving the 13 bp element in a semi-autonomous manner, as has been recently shown to involve direct promoter occupancy and competition with the activating transcription factor NF-Y[15,25]. In addition, we also demonstrate a key role for distal long-range interactions involving a functional region upstream of the *HBD* gene that enables effective HbF silencing[17]. While this *HBD* upstream region has been posited to require BCL11A, originally through the suggested occupancy of this factor in this region, it has been unclear whether this region is functionally required for HbF silencing. Through our functional assays in single progenitors and CAPTURE data, we show that long-range chromatin organization involving this region requires intact BCL11A activity. While the precise underlying mechanisms that account for this distal activity remain to be fully defined, particularly given the variable observed binding from recent CUT&RUN or chromatin immunoprecipitation–sequencing studies of BCL11A in this region[15,25] (Supplementary Fig. 1b), a key role for BCL11A activity is clearly present. Importantly, we also demonstrate distinct and non-overlapping roles for ZBTB7A in HbF silencing and a surprisingly strong negative interaction between its perturbation and the *HBB*-3.5kb deletion. Collectively, our findings lead to a unified model of how distinct mutations may in aggregate contribute to the silencing of HbF and how BCL11A has dual activities to effectively silence HbF through both local promoter interactions and long-range distal modulation of chromatin conformation at this locus (Fig. 5d, e). An important area for future studies will be to tease apart whether overlapping or separable activities of BCL11A are required for these distinct roles involved in regulating HbF. A deeper understanding of this process holds significant promise for the development of improved and more effective therapeutic approaches to induce HbF in sickle cell disease and β-thalassemia. Moreover, such complex long-range interactions suggest a reason why only select approaches can

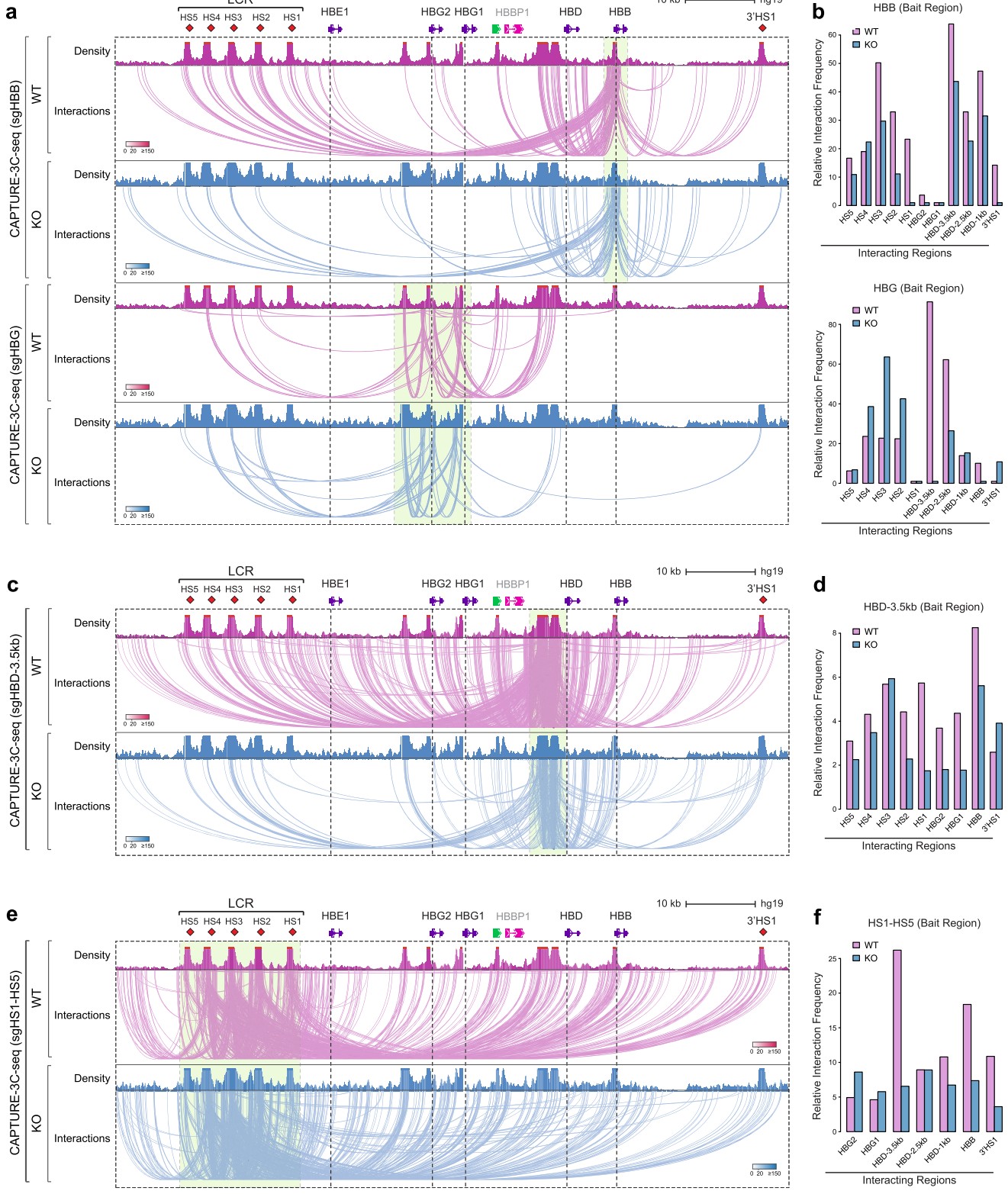

**Fig. 4 Analysis of distinct β-globin locus *cis*-regulatory elements and BCL11A-dependent interactions using CAPTURE. a**, **c**, **e** Genome browser view of long-range DNA interactions in WT (purple) vs BCL11A KO (blue) HUDEP-2 cells using *HBB*, *HBG* (**a**), *HBD-HBG* intergenic region (**c**), and LCR (HS1-HS5) (**e**) as the capture bait region, respectively. **b**, **d**, **f** Quantitative analysis of relative interaction frequency between the captured bait and the interacting regions. Three anchor sites in the *HBD*-3.5kb interval region are shown as −3.5, −2.5, and −1 kb sites upstream of *HBD*.

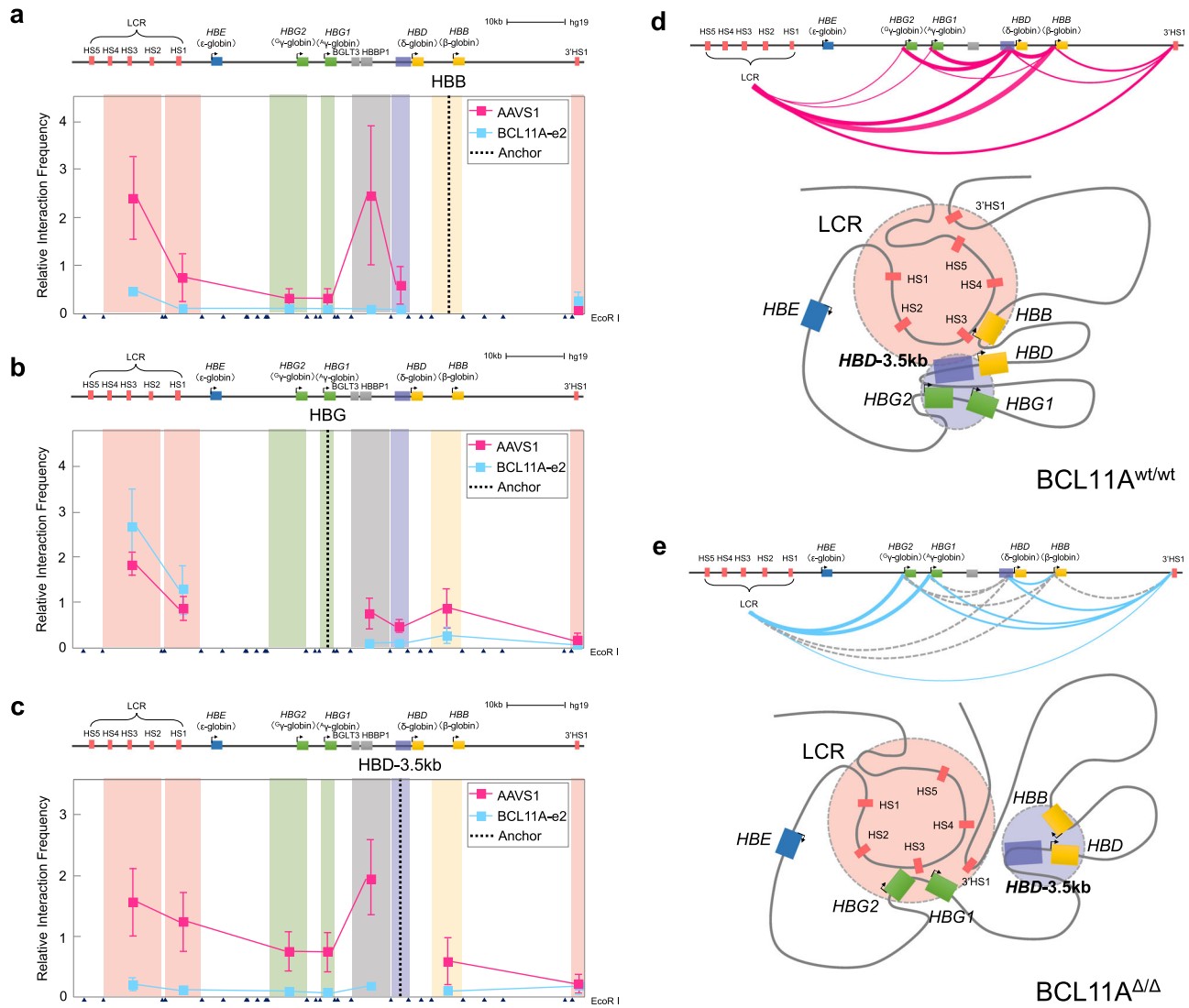

**Fig. 5 Chromosome conformation capture analysis and a model for regulation of the β-globin locus by BCL11A to alter HbF expression.** Chromosome conformation capture (3C) analysis was performed for HSPC-derived cell nuclei (AAVS1 and BCL11A exon 2 gRNA) at day 12 of erythroid differentiation. EcoRI cutting sites are shown at the bottom of each panel by triangles, and the designated anchors are plotted using dotted vertical lines. Relative interaction frequencies are shown for AAVS1 control in pink and BCL11A exon 2 editing in blue. Error bars indicate SEM from three biological replicates. **a** Relative interaction frequency between the *HBB* fragment as anchor and other regions of the β-globin locus. **b** Relative interaction frequency between the *HBG* fragment as anchor and other regions of the β-globin locus. **c** Relative interaction frequency between the *HBD*-3.5kb fragment as anchor and other regions of β-globin locus. **d**, **e** illustrate major long-range interactions at the top with a model of how these findings translate to a model of chromatin conformation at the locus. Purple line: chromatin interaction in HUDEP-2 WT; blue line: chromatin interaction in BCL11A KO; gray dotted line: chromatin interaction loss in BCL11A KO compared to WT. The width of each line indicates the strength of chromatin interaction observed.

successfully reconfigure chromatin interactions and gene expression at this locus[33].

While we have gained important insights through our analysis, there are also several limitations that should be noted about these results. First, we have focused on mimicking perturbations highlighted by rare in vivo observations in humans. Our rationale was based on the fact that these particular elements have been definitively shown to have a significant role in HbF silencing in humans. While other factors and elements have been suggested to be involved in this process, we have not examined these here, since our focus was to understand how various in vivo perturbations may cooperatively function in the process of HbF silencing. For instance, a set of long-range interactions with a region upstream of the *HBD*-3.5kb element within the *HBBP1* pseudo-gene and also involving the *BGLT3* non-coding RNA have been suggested to have roles in HbF silencing[36,37]. Future studies

taking advantage of similar kinds of approaches as we describe here will enable further insights and provide an opportunity to examine the role of other *trans*- and *cis*-acting regulators through perturbation in single cells. Second, we do note limitations of the single progenitor functional assay we describe. Most notably, the baseline level of *HBG1/2* mRNA expression observed in this assay is higher than what is observed in vivo. This reinforces our focus on perturbations where the effects can be compared to what is seen in vivo in the setting of individual perturbations. While the resultant levels of individual perturbations were larger than what is observed in vivo in humans, the overall degree of HbF induction does closely mimic what is observed in individuals with these rare mutations, providing confidence in this approach. Third, we are constrained by specific genome edits that can readily be achieved through existing genome-editing tools. This has limited our ability to combinatorially target in *cis* multiple

regulatory elements within the β-globin locus. As improved genome perturbation and editing tools are developed, such as base and prime editors[38], it is likely that refined assessment of distinct perturbations, including the enumerable SNVs linked to this process, can be uncovered.

Finally, our findings also have broader implications for how we should consider the multifactorial impact of genetic variation. We have traditionally focused on the impact of individual variants either uncovered through studies of rare diseases or complex traits. This has enabled previously unappreciated knowledge and emphasized the key role of non-coding regulatory variation. However, many questions remain about how numerous variants collectively act to impact a phenotype of interest[39,40]. Through our studies of HbF regulation—a simple and directly measured quantitative trait impacted by human genetic variation—we have shown how studies of combinatorial variation can reveal synergistic interactions that move beyond the conventional additive models employed to study human genetic variation. As increasing numbers of common variants underlying complex diseases are uncovered, approaches combining single-cell functional interrogation and locus-specific chromatin interaction assays, as we use here, are likely to provide insights into how combinations of genetic variants can act in non-additive ways to modulate the 3D genome in human health and disease[41,42].

## Methods

**Primary HSPC culture and colony-forming cell assay.** Human HSPCs from healthy donors (CD34 enriched) were obtained from the Fred Hutchinson Hematopoietic Cell Processing and Repository (Seattle, USA). Use of deidentified human HSPCs was approved by the Institutional Review Board at Boston Children's Hospital. The HSPCs were thawed and cultured in phase I erythroid differentiation Medium (EDM) composed of Iscove's modified Dulbecco's medium (IMDM, Life Technologies) supplemented with 200 μg/mL human holo-transferrin, 10 μg/mL recombinant human insulin, 3 IU/mL heparin, 2% human AB plasma, 3% human AB serum, and 1% penicillin/streptomycin) with three supplemental cytokines (3 IU/mL erythropoietin (EPO), 10 ng/mL stem cell factor (SCF), 1 ng/mL interleukin-3 (IL-3)) at 37 °C and 5% $CO_2$[43–45]. For colony-forming cell assays, HSPCs were plated at 500 cells/mL in MethoCult H4034 Optimum methylcellulose medium (StemCell Technologies, Inc.) that support erythroid maturation, along with formation of other hematopoietic colony types. Individual BFU-E colonies were identified by the distinct red color and morphology, which separates these cells from other hematopoietic colonies. The cells in the BFU-E colonies were picked at day 14 after plating.

**CRISPR/Cas9 genome editing in HSPCs.** gRNA sequences (shown in Supplementary Data 3) were chosen using the IDT CRISPR design tool and generated as oligonucleotides. The Amaxa Nucleofector System with program DZ-100 was used to deliver Cas9 and gRNA as a ribonucleoprotein (RNP) complex into HSPCs. In all, 5 μL of 120 pmol gRNA duplex (crRNA:tracrRNA) and 105 pmol Cas9 protein (Integrated DNA Technologies, following the manufacturer's instructions) were prepared to form RNP complexes and delivered in 20 μL of $2–4 × 10^5$ HSPCs using the P3 Primary Cell 4D Nucleofector X Kit S (Lonza, following the manufacturer's instructions). Subsequently, cells were transferred to phase I EDM culture medium for 24 h and then seeded in MethoCult H4034 Optimum methylcellulose medium (StemCell Technologies, Inc.). For long-range deletions (HBB-3.5kb, HBB-HBD, HBD-3.5kb), paired gRNAs were delivered into HSPC following colony-forming cell assay. To generate combinatorial perturbations of cis-regulatory regions (HBB-3.5kb, HBG1/2-Δ13bp) and trans-acting factors (BCL11A, ZBTB7A), multiplexed RNPs composing 2–3 gRNAs were simultaneously nucleofected into HSPCs. BFU-E colonies were randomly selected and collected as a single group of cells after 2 weeks of culture in order to evaluate the genome-editing genotype and hemoglobin gene expression. DNA and RNA were simultaneously isolated from single BFU-E colony using the ALLPrep DNA/RNA Micro Kit (Qiagen), following the manufacturer's protocol. cDNA was synthesized from BFU-E colony-extracted RNA using the iScript cDNA Synthesis Kit (BioRad) according to the manufacturer's instructions. Primer sequences used for PCR screening and Sanger sequencing of gRNA editing efficiency are shown in Supplementary Data 4.

**Detection of genome-editing events.** To evaluate editing efficiency of gRNA, we performed PCR followed by Sanger sequencing and ICE Analysis (https://ice.synthego.com/) to decompose insertion/deletion (indel) of editing outcomes. Genotypes of edited alleles were confirmed by comparing the sequence chromatogram from edited bulk HSPCs to a control sequence chromatogram from WT HSPCs. These genotypes were also confirmed by independent massively parallel sequencing

validation. Quantitative PCR (qPCR) was performed using SYBR green (BioRad) and primers (shown in Supplementary Data 5) to evaluate the frequency of deletion and inversion of target regions (HBB-3.5kb, HBB-HBD, HBD-3.5kb) and was validated with deletions confirmed via orthogonal approaches.

For genotyping of BFU-E colonies, CRISPR/Cas9 targeted loci (BCL11A exon 2 gRNA, BCL11A exon 4 gRNA, ZBTB7A gRNA, HBG1/2 13 bp gRNA) were amplified using primers (shown in Supplementary Data 4) and the Q5 High-Fidelity 2X Master Mix (New England Biolabs) according to the protocol. Amplicons were sequenced for genotyping using primers (shown in Supplementary Data 4). The intact HBG1/2 genetic loci were confirmed by qPCR using primers (shown in Supplementary Data 5). Mutation of >2 bp within HBG-Δ13bp were considered disruptive mutations under the assumption that these edits overlapped with the characterized BCL11A-binding site. The deletion and inversion events (HBB-3.5kb, HBB-HBD, HBD-3.5kb) were identified by qPCR using primers. qPCR primer sequences used for detection of genome-editing events are shown in Supplementary Data 5.

**Globin gene expression analysis.** For cDNA extracted from BFU-E colonies, qPCR was carried out using a 96-well plate on a CFX96 Real Time System (BioRad) with SYBR Green Supermix (BioRad). HBB and HBG1/2 mRNA/cDNA expression levels were normalized with endogenous control HBA1/2 gene expression levels. Gene-specific primers used for qPCR are listed in Supplementary Data 6. HBG1 and HBG2 gene expression were identified by HBG1/2 expression level and HBG1:HBG2 expression ratio, which were calculated based on G (HBG1):A (HBG2) nucleotide ratios from the Sanger sequencing chromatograms. PCR and sequencing primers are shown in Supplementary Data 6.

**Erythroid differentiation.** HSPCs were cultured and differentiated in a three-phase erythroid differentiation culture system[43–45]. Briefly, cells were cultured in IMDM (Life Technologies) containing 200 μg/mL human holo-transferrin, 10 μg/mL recombinant human insulin, 3 IU/mL heparin, 2% human AB plasma, 3% human AB serum, and 1% penicillin/streptomycin (base medium). In phase I (0–7 days), the cultures were supplemented with 3 IU/mL EPO, 10 ng/mL SCF, and 1 ng/mL IL-3 and in phase II (7–12 days) they were supplemented with 3 IU/mL EPO and 10 ng/mL SCF alone. In phase III (12–17 days), primary cell cultures contained 1 mg/mL of human holo-transferrin supplemented with 3 IU/mL EPO. Cells were maintained at $10^5–10^6$ per mL in phases I and II and at $1–5 × 10^6$ per mL in phase III. Cells were changed into fresh culture medium every 3 days. For assessment of differentiation, cells were washed in phosphate-buffered saline (PBS) and stained with anti-human CD49d, CD71, and CD235a antibodies (shown in Supplementary Data 10). Flow cytometric analyses were conducted on an Accuri C6 instrument and all data were analyzed using the FlowJo software (v.10.3).

**Immunoblot analysis.** Immunoblotting was performed by following standard approaches[45]. Cells were lysed in RIPA buffer (Santa Cruz Biotechnology), and protein levels were quantified with the DC Protein Assay (BioRad). Briefly, samples were incubated at 95 °C for 5 min in 4× Laemmli sample buffer (BioRad) and loaded on to a Mini-Protein TGX Gel (BioRad) for electrophoresis at 120 V for 1 h. Proteins were then transferred to polyvinylidene difluoride membrane with BioRad wet transfer system at 100 V for 1.5 h. Membranes were blocked with TBS-T/3% bovine serum albumin (BSA) for 1 h and then incubated with primary antibodies overnight in a cold room with shaking. Excess antibodies were washed with TBS-T (50 mM Tris pH 8.0, 150 mM NaCl, 1% Tween 20) for 3 times and horseradish peroxidase-conjugated secondary antibodies were incubated for 30 min at room temperature. After 3 washes with TBS-T, the membranes were developed with the Clarity Western ECL Substrate Kit (BioRad). Immunoblots were performed with antibodies BCL11A (AbCam, at 1/1000 dilution), ZBTB7A (eBioscience, at 1/1000 dilution), Hemoglobin γ (Santa Cruz, 1/250 dilution), and glyceraldehyde-3-phosphate dehydrogenase (Santa Cruz, at 1/5000 dilution). The secondary antibodies used were goat anti-rabbit (BioRad), goat anti-mouse (BioRad), and goat anti-hamster (Thermo Fisher Scientific). Antibodies are listed in Supplementary Data 9. The blotting intensities were analyzed using ImageJ (version 1.8.0).

**RNA isolation and qPCR with reverse transcription (RT-qPCR).** In all, $1 × 10^6$ HSPC-derived erythroblasts were collected and isolated for RNA using the RNeasy Mini Kit (Qiagen) with on-column DNAse (Qiagen) digestion, according to the manufacturer's instructions. cDNA was synthesized with the iScript cDNA Synthesis Kit (BioRad). RT-qPCR was carried out using a 96-well plate on a CFX96 Real Time System (BioRad) with SYBR Green Supermix (BioRad). Gene-specific primers used for RT-qPCR are listed in Supplementary Data 6.

**HbF flow cytometry.** For HbF analysis cells were fixed in 0.05% glutaraldehyde for 10 min, washed 2 times with PBS/0.1% BSA (Sigma), and permeabilized with 0.1% Triton X-100 (Life Technologies, prepared in PBS/0.1% BSA) for 5 min. Following one wash with PBS/0.1% BSA, cells were stained with HbF-APC conjugate antibody (Invitrogen). For primary erythroid cells, $5 × 10^5$ cells were incubated with 0.4 μg HbF-APC antibody (shown in Supplementary Data 10) for 10 min in the dark at room temperature. Cells were then washed twice with PBS/0.1% BSA. Flow

cytometric analyses were conducted on an Accuri C6 instrument, and all data were analyzed using the FlowJo software (v.10.3).

**Hemoglobin HPLC.** Approximately $5 \times 10^6$ erythroblasts were collected on day 17 of erythroid differentiation and subjected to lysis for hemoglobin high-performance liquid chromatography (HPLC). HbF levels were measured using a G7 HPLC Analyzer (Tosoh Bioscience, Inc.) with the β-thalassemia program. To compare HbF levels across experiments, we defined HbF% as a proportion of HbF relative to HbA and other hemoglobin subtypes.

**HUDEP-2 cell culture.** HUDEP-2 cells were maintained in expansion medium: StemSpan SFEM (STEMCELL Technologies) with 50 ng/mL SCF, 3 IU/mL EPO, $10^{-6}$ M dexamethasone, and 1 μg/mL doxycycline and passaged every 3 days. The cell density was maintained within 20,000–500,000 cells/mL. Erythroid differentiation was carried out by replacing the medium to EDM (IMDM; Corning) supplemented with 330 μg/mL human holo-transferrin, 10 μg/mL recombinant human insulin, 2 IU/mL heparin, 5% heat-inactivated plasma, 3 IU/mL EPO, 2 mM L-glutamine) with two supplements (100 ng/mL SCF, 1 μg/mL doxycycline). After 5 days of differentiation, cells were collected for CAPTURE analyses.

**Biotinylated dCas9-mediated CAPTURE assays.** HUDEP-2 cells co-expressing Bio-tagged dCas9 (CAPTURE2.0-CBio) and sequence-specific sgRNAs were generated as previously described with modifications[21,22]. CAPTURE-3C-seq experiments were performed as previously described using between 3 and 5 million WT or BCL11A knockout HUDEP-2 cells[21,22]. CAPTURE-3C-seq data processing and quantitative analyses of locus-specific long-range DNA interactions were performed as previously described[21,22]. Three biological replicates were merged for WT or BCL11A knockout HUDEP-2 cells, respectively. Only significant long-range DNA interactions from the following captured bait regions with BF score ≥20 were used for quantitative analyses: HBB, HBG1/2, HBD-3.5kb (containing sgRNAs targeting HBD-1kb, HBD-1.5kb, and HBD-3.5kb regions), and upstream enhancer locus control (HS1–5) regions. The relative interaction frequency per kilobase at each captured bait region was calculated as (interactions between bait region and other interacting regions) $\times 10^6$/(size of bait region × all interactions from bait region). The computer code for data processing and analysis is available from GitHub (https://github.com/ChenYong-RU/MAXIM).

**Chromosome conformation capture.** 3C was performed as described previously[46]. Briefly, $1 \times 10^7$ HSPC-derived cell nuclei (AAVS1 and BCL11A exon 2 gRNA) at day 12 of erythroid differentiation were fixed in 1% formaldehyde, digested with EcoRI overnight, and ligated for 4 h with T4 DNA ligase at 16 °C. Cross-links were reversed and ligation products extensively purified. qPCR was performed using SYBR green (BioRad) and primers (shown in Supplementary Data 7) to evaluate ligation products. Ligation frequencies were normalized to an interaction in the TUBA1A gene (shown in Supplementary Data 7).

**Creation of BCL11A dominant-negative mutants.** All constructs of BCL11A dominant-negative mutants incorporating 1 bpΔ, 2 bpΔ, and 7 bpΔ mutants of interest were created using the Q5 Site-Directed Mutagenesis Kit (New England Biolabs). Primers used for constructs are listed in Supplementary Data 8. For genotyping of BCL11A dominant-negative mutants, targeted loci were amplified using primers (shown in Supplementary Data 8) and the Q5 High-Fidelity 2× Master Mix (New England Biolabs) according to the protocol. Amplicons were sequenced for genotyping using primers (shown in Supplementary Data 8).

**Lentiviral infections.** HSPCs undergoing erythroid differentiation were transduced with HMD control, BCL11A WT, and 1 bpΔ, 2 bpΔ, and 7 bpΔ lentiviral constructs on day 2 of differentiation; 293T cells for lentivirus production were cultured in DMEM (Life Technologies) with 10% fetal bovine serum and 1% penicillin/streptomycin. Approximately 24 h before transfection, 293T cells were seeded, without antibiotics, in 6-well plates. Cells were co-transfected with the packaging vectors pVSVG and pΔ8.9 and the lentiviral genomic vector of interest. The medium was changed to base medium on the day following transfection, and the viral supernatant was collected approximately 48 h post-transfection, filtered with a 0.45-μm filter, and used for infection of CD34+ cells. Where the virus was required to be concentrated, 293T cells were seeded in 10-cm dishes and the viral supernatant was filtered and centrifuged at 64,512 × g for 2 h at 4 °C. Between 200,000 and 300,000 CD34+ cells were infected in 6-well plates with 8 μg/mL of polybrene (Millipore), spun at 448 × g for 1.5 h at room temperature, and incubated in the viral supernatant overnight at 37 °C. Virus was washed off 1 day after infection, and infected cells were selected for by green fluorescent protein (GFP) expression driven by IRES-GFP in the HMD vector in the particular construct. GFP+ cells were sorted by fluorescence-activated cell sorting in a sterile manner and cultured for further analysis. Flow cytometric analyses were conducted on Becton Dickinson LSRII, and data were analyzed using the FlowJo software (v.10.3).

**Statistical analyses and quantitative modeling.** Data were analyzed using Microsoft Excel (version 16.16), GraphPad Prism 7, FlowJo (version 10.3), ImageJ (version 1.8.0), and R (version 3.4.3). Statistical analysis of data was generally done using the two-tailed Student's $t$ test, unless otherwise specified. Detailed information about statistical methods are specified in figure legends or in specific parts of the "Methods" section.

In order to quantitatively model HbF expression, we separated this measurement into HBB and HBG1/2 mRNA expression and fit these values using a LMM. Edited genotypes were coded as either editing events (0–2) for the HBB-HBD, HBD-3.5kb, the combined deletion HBB-3.5kb, and ZBTB7A or as an allelic series for HBG1/2 (0–4) and BCL11A (0–3) and treated as fixed effects in these models. Interaction effects between the edits were also treated as fixed effects. Three models with differing fixed effects were used in order to maximize numeric observations and to explore meaningful genetic interactions: (1) HBB-HBD, HBD-3.5kb, and their interaction; (2) HBB-3.5kb, HBG1/2, and BCL11A with applicable interaction terms; and (3) HBB-3.5kb, ZBTB7A, and BCL11A and applicable interaction terms. In order to account for differences in the unedited states between conditions, we allowed for random intercepts based on the gRNAs introduced in each condition, as well as the specific cell donor when applicable. Thus, the LMMs fit the equation:

$$y = X\beta + Z\mu + \epsilon \quad (1)$$

where $y$ is $n \times 1$ column vector of HBB or HBG1/2 mRNA measurements, $X$ is a $n \times$ number of deletions + interaction terms matrix of fixed effects ($p$), $\beta$ is a $p \times 1$ row vector of the $\beta$-coefficients of the fixed effects, $Z$ is an $n \times$ number of donors + number of gRNAs, which affect either HBB or HBG1/2 basal levels (doubled for the presence or absence of the gRNA) model matrix of random effects, and $\epsilon$ is $n \times 1$ residual column vector of the residuals.

LMM was performed using the R package lme4. $P$ values for fixed effects were approximated using the Satterthwaite method instantiated through the lmerTest package. We do note that large confidence intervals were noted in some cases, particularly at the end of the allelic spectrum involving multiple edits/perturbations due to limited numbers of colonies of a particular genotype assessed. All resultant $\beta$ and corresponding $P$ values are provided in the text as each analysis is discussed. Data were visualized in R using the sjPlot and ggplot2 packages.

**Reporting summary.** Further information on research design is available in the Nature Research Reporting Summary linked to this article.

## Data availability

The original data for globin gene expression analysis in BFU-E has been provided as a Supplementary File. Raw immunoblots for Supplementary Figs. 7b, 8d, and 10b are available online. CAPTURE data are available at the Gene Expression Omnibus (https://www.ncbi.nlm.nih.gov/geo/) under accession number GSE181151. All previously published datasets used in this paper are available at the Sequence Read Archive (https://www.ncbi.nlm.nih.gov/sra) and relevant accession numbers are listed in Supplementary Data 2. Source data are provided with this paper.

## Code availability

Custom computational code for reproduction of statistical analyses and quantitative modeling is available at https://github.com/sankaranlab/Fetal_Hemoglobin_model.

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

## Acknowledgements

We are grateful to members of the Sankaran laboratory for valuable guidance and suggestions. We thank R. Rosales, G. Menard, and D. Dorfman for assistance with hemoglobin HPLC analyses. This work was supported by the New York Stem Cell Foundation (to V.G.S.), a gift from the Lodish Family to Boston Children's Hospital (to V.G.S.), and National Institutes of Health Grants R01 DK103794 (to V.G.S.), R01 HL146500 (to V.G.S.), R56 DK125234 (to V.G.S.), R01CA230631 (to J.X.), and R01DK111430 (to J.X.). J.X. is a Scholar of The Leukemia & Lymphoma Society. S.K.N. and J.X. are Scholars of the American Society of Hematology. V.G.S. is a New York Stem Cell Foundation-Robertson Investigator.

## Author contributions

Y.S. and V.G.S. conceptualized the study. Y.S., J.M.V., N.L. Y.Z., Y.J.K., J.X., and V.G.S. developed methodology. Y.S., J.M.V., N.L., Y.Z., Y.J.K., S.M., and A.E. performed experiments. Y.S., J.M.V., N.L., Y.Z., Y.J.K., J.X., and V.G.S. analyzed experiments. Y.S., J.M.V., N.L., Y.Z., Y.J.K., S.K.N., R.A.V., C.F., A.B., S.H.O., J.X., and V.G.S. developed resources. Y.S., J.M.V., Y.Z., N.L., J.X., and V.G.S. wrote the original draft with input from all authors. S.H.O., J.X., and V.G.S. supervised the project.

## Competing interests

The authors declare no competing interests.
