## [Peer Review File · Nature Communications]

REVIEWER COMMENTS

Reviewer #1 (Remarks to the Author):

This manuscript describes an exciting approach to dissecting complex mechanisms of gene regulation using directed genome edits (single and multiple) in single cells (HSPCs) and analyzing the functional impacts of those mutations in clonal populations (BFU-E) derived from the single edited cells. The studies are conducted in the context of fetal hemoglobin (HbF) regulation in erythroid cells, for which multiple strong candidates for both cis-regulatory elements and trans-acting factors (especially BCL11A and ZBTB7A) have been identified from decades of genetic and biochemical investigation. The authors focused on well-known elements and proteins identified by rare mutations in humans. They showed that their approach can recapitulate the known effects of cis-acting regulatory elements located proximal to the HBG1&2 genes encoding fetal gamma-globins (specifically the 13bp element in promoters) and distal (a 3.5kb region upstream of the HBD gene encoding delta-globin), as well as the effects of knocking out the major repressors of the HBG1&2 genes, BCL11A and ZBTB7A. With that foundation, they were able to explore interactions between these regulatory elements and BCL11A by generating allelic series for the regulatory elements and BCL11A gene using multiplexed guide RNAs to introduce multiple, directed mutations in the single HSPCs. Analysis of expression of HBG1&2 and HBB in the clones (isogenic progeny of the edited HSPCs) led to several insights. These discoveries include clear evidence for a functional interaction between the proximal 13bp regulatory element in the promoters of HBG1&2 with BCL11A, as well as evidence for a connection between the distal -3.5 region of HBD and BCL11A. Previous evidence had suggested that these interactions and connections may occur, but the present study provides clear genetic evidence to support them. I should stress that these issues have been widely discussed within the community with diverging views, and the data in this manuscript go a long way to resolving some of these issues. The authors also conduct a focused chromatin interaction study of the impact of loss of the BCL11A protein on interactions of these several cis regulatory elements with other elements and with the HBG1&2 and HBB genes, using the CAPTURE approach. These results support the conclusions from the genetic analyses and lead to an informative model that integrates changes in the 3D architecture of the HBB locus with shifts in gene expression. The presentation is clear, and limitations are stated explicitly.

Over previous rounds of review, issues raised in earlier versions have been addressed. I raise one remaining issue and some additional comments.

(1) It is not clear what aspects of the results in this manuscript show that ZBTB7A is acting only locally, in contrast to BCL11A, which the authors show is connected genetically both proximally (the 13bp promoter deletion) and distally (the HBB-3.5kb deletion) to regulation of the HBG genes. The evidence for these genetic connections for BCL11A is described as "antagonistic interactions" revealed by the linear mixed model (lines 261-262 for the 13bp deletion, lines 277-279 for HBB-3.5kb deletion). In particular, a key observation is that the slope of the line in the interaction plots

(for predicted levels of HBG1&2 RNA vs. increasing numbers of deletions of the cis-elements) decreased over an allelic series of decreased BCL11A (Fig. 2, c and f). When a similar experiment is done with an allelic series for the ZBTB7A gene, a similar decrease in slope of the interaction plot (HBG1&2 RNA vs increasing numbers of deletions of HBB-3.5kb deletion) is observed as the level of ZBTB7A decreased (Fig. 3c). The authors describe this result as a "significant attenuation of the HBG1/2 induction ..." (lines 288-290). This result appears to show that the number of copies of the distal HBB-3.5kb region does influence the repressive activity of ZBTB7A on expression of the HBG genes. However, the authors draw a more complicated conclusion that is hard to interpret, i.e. "ZBTB7A's local role in silencing the HBG1/2 genes intersected with pathways that enable HbF induction upon long-range perturbations" (lines 292-293). It would seem simpler and more consistent to conclude that there is evidence for a role for long-range, distal interactions as well as local interactions in the role of ZBTB7A in repressing expression of the HBG genes. Furthermore, the ZBTB7A binding data shows evidence of binding in the HBB-3.5kb region (Extended data Fig. 1).

(2) In showing that BCL11A and ZBTB7A are acting independently in this assay, it appears that the production of parallel lines in the interaction plots (Fig. 3f) was an important observation. The authors may want to make this point explicitly in the text to make the case and clarify interpretation of the interaction plots for readers.

(3) Lines 401-402: The authors describe a "lack of observable binding from recent CUT&RUN or CHIP-seq studies of BCL11A" in the HBB-3.5kb region. However, Extended Data Fig. 1b does show a notable signal for BCL11A occupancy in this region when assayed by CUT&RUN, whereas the signal assayed by CHIP-seq appears similar to background. Perhaps the authors want to state that evidence for binding by BCL11A in this region is inconsistent among recent reports with different assays.

In the interest of transparency, I sign the review.

Ross Hardison

Reviewer #2 (Remarks to the Author):

The Authors have answered to several of the raised points. However, some points remained not fully addressed:

- I tried to clarify my previous question: Why in LRF KO clones harboring heterozygous deletion of the HBB 3.5 region, gamma-globin expression is reduced to levels similar to the ones observed in clones harboring homozygous deletions of the HBB 3.5 region? If LRF has a local role, I would have expected still higher gamma globin levels in heterozygous clones (harboring heterozygous deletion of the HBB 3.5 region).

- Thalassemia phenotype: I am not sure that the Authors could talk about a thalassemic phenotype (e.g., in Figure 1d) only looking at RNA expression and without further analyses: 50% of beta-like globin transcripts could be comparable to the situation observed in a beta-thalassemia carrier

- If the Authors use globin RNA expression as an indicator of a beta-thalassemia phenotype, then the excess in beta-like globin expression observed in Figure 3b (LRF KO) should be considered as causing an alpha thalassemic phenotype, can the Author comment on that? Such a high gamma-globin expression in LRF KO clones is interesting, I think analyses of the expression of erythroid markers by qRT-PCR could be useful to confirm that these data are not biased by the potential different developmental/hemoglobinization stage of LRF KO clones.

- I would tone down the conclusions of the Capture C experiments

REVIEWER COMMENTS

Reviewer #1 (Remarks to the Author):

This manuscript describes an exciting approach to dissecting complex mechanisms of gene regulation using directed genome edits (single and multiple) in single cells (HSPCs) and analyzing the functional impacts of those mutations in clonal populations (BFU-E) derived from the single edited cells. The studies are conducted in the context of fetal hemoglobin (HbF) regulation in erythroid cells, for which multiple strong candidates for both cis-regulatory elements and trans-acting factors (especially BCL11A and ZBTB7A) have been identified from decades of genetic and biochemical investigation. The authors focused on well-known elements and proteins identified by rare mutations in humans. They showed that their approach can recapitulate the known effects of cis-acting regulatory elements located proximal to the HBG1&2 genes encoding fetal gamma-globins (specifically the 13bp element in promoters) and distal (a 3.5kb region upstream of the HBD gene encoding delta-globin), as well as the effects of knocking out the major repressors of the HBG1&2 genes, BCL11A and ZBTB7A. With that foundation, they were able to explore interactions between these regulatory elements and BCL11A by generating allelic series for the regulatory elements and BCL11A gene using multiplexed guide RNAs to introduce multiple, directed mutations in the single HSPCs. Analysis of expression of HBG1&2 and HBB in the clones (isogenic progeny of the edited HSPCs) led to several insights. These discoveries include clear evidence for a functional interaction between the proximal 13bp regulatory element in the promoters of HBG1&2 with BCL11A, as well as evidence for a connection between the distal -3.5 region of HBD and BCL11A. Previous evidence had suggested that these interactions and connections may occur, but the present study provides clear genetic evidence to support them. I should stress that these issues have been widely discussed within the community with diverging views, and the data in this manuscript go a long way to resolving some of these issues. The authors also conduct a focused chromatin interaction study of the impact of loss of the BCL11A protein on interactions of these several cis regulatory elements with other elements and with the HBG1&2 and HBB genes, using the CAPTURE approach. These results support the conclusions from the genetic analyses and lead to an informative model that integrates changes in the 3D architecture of the HBB locus with shifts in gene expression. The presentation is clear, and limitations are stated explicitly.

We thank the Reviewer for their supportive comments on our work and appreciate their discussing how we provide “clear genetic evidence” to support the interactions we study here.

Over previous rounds of review, issues raised in earlier versions have been addressed. I raise one remaining issue and some additional comments.

(1) It is not clear what aspects of the results in this manuscript show that ZBTB7A is acting only locally, in contrast to BCL11A, which the authors show is connected genetically both proximally (the 13bp promoter deletion) and distally (the HBB-3.5kb deletion) to regulation of the HBG genes. The evidence for these genetic connections for BCL11A is described as "antagonistic

interactions" revealed by the linear mixed model (lines 261-262 for the 13bp deletion, lines 277-279 for HBB-3.5kb deletion). In particular, a key observation is that the slope of the line in the interaction plots (for predicted levels of HBG1&2 RNA vs. increasing numbers of deletions of the cis-elements) decreased over an allelic series of decreased BCL11A (Fig. 2, c and f). When a similar experiment is done with an allelic series for the ZBTB7A gene, a similar decrease in slope of the interaction plot (HBG1&2 RNA vs increasing numbers of deletions of HBB-3.5kb deletion) is observed as the level of ZBTB7A decreased (Fig. 3c). The authors describe this result as a "significant attenuation of the HBG1/2 induction ..." (lines 288-290). This result appears to show that the number of copies of the distal HBB-3.5kb region does influence the repressive activity of ZBTB7A on expression of the HBG genes. However, the authors draw a more complicated conclusion that is hard to interpret, i.e. "ZBTB7A's local role in silencing the HBG1/2 genes intersected with pathways that enable HbF induction upon long-range perturbations" (lines 292-293). It would seem simpler and more consistent to conclude that there is evidence for a role for long-range, distal interactions as well as local interactions in the role of ZBTB7A in repressing expression of the HBG genes. Furthermore, the ZBTB7A binding data shows evidence of binding in the HBB-3.5kb region (Extended data Fig. 1).

We appreciate this valuable suggestion and completely agree that this simpler model is most consistent with the data. We have now clarified this in the manuscript and state, "This finding demonstrates that ZBTB7A also acts through both local and long-range interactions to silence HbF."

(2) In showing that BCL11A and ZBTB7A are acting independently in this assay, it appears that the production of parallel lines in the interaction plots (Fig. 3f) was an important observation. The authors may want to make this point explicitly in the text to make the case and clarify interpretation of the interaction plots for readers.

We are grateful for this suggestion and have now more clearly mentioned this in the text.

(3) Lines 401-402: The authors describe a "lack of observable binding from recent CUT&RUN or ChIP-seq studies of BCL11A" in the HBB-3.5kb region. However, Extended Data Fig. 1b does show a notable signal for BCL11A occupancy in this region when assayed by CUT&RUN, whereas the signal assayed by ChIP-seq appears similar to background. Perhaps the authors want to state that evidence for binding by BCL11A in this region is inconsistent among recent reports with different assays.

We appreciate this suggestion and have now more clearly discussed this in the manuscript.

In the interest of transparency, I sign the review.

Ross Hardison

Reviewer #2 (Remarks to the Author):

The Authors have answered to several of the raised points. However, some points remained not fully addressed:

- I tried to clarify my previous question: Why in LRF KO clones harboring heterozygous deletion of the HBB 3.5 region, gamma-globin expression is reduced to levels similar to the ones observed in clones harboring homozygous deletions of the HBB 3.5 region? If LRF has a local role, I would have expected still higher gamma globin levels in heterozygous clones (harboring heterozygous deletion of the HBB 3.5 region).

This Reviewer raises an excellent and important point. One issue is that the responses may not be linear. Some expression of ZBTB7A may be sufficient to suppress *HBG1/2* mRNA outside of physiologic levels. However, we would note that there is indeed an elevation observed, but this is minor in comparison to the effect seen with homozygous/ compound heterozygous edits of ZBTB7A. We have modified the discussions to attempt to more clearly discuss this.

- Thalassemia phenotype: I am not sure that the Authors could talk about a thalassemic phenotype (e.g., in Figure 1d) only looking at RNA expression and without further analyses: 50% of beta-like globin transcripts could be comparable to the situation observed in a beta-thalassemia carrier

We appreciate this comment and have modified our statement to more clearly discuss beta-like to alpha-like mRNA imbalance, rather than using the ambiguous term thalassemia phenotype.

- If the Authors use globin RNA expression as an indicator of a beta-thalassemia phenotype, then the excess in beta-like globin expression observed in Figure 3b (LRF KO) should be considered as causing an alpha thalassemic phenotype, can the Author comment on that? Such a high gamma-globin expression in LRF KO clones is interesting, I think analyses of the expression of erythroid markers by qRT-PCR could be useful to confirm that these data are not biased by the potential different developmental/hemoglobinization stage of LRF KO clones.

We are now no longer using the term thalassemia phenotype and specifically discussing findings on globin mRNA imbalance to more precisely describe our findings.

- I would tone down the conclusions of the Capture C experiments

We appreciate this suggestion. We have now modified the text to tone down these conclusions and ensure that they only “suggest” potential mechanisms.

REVIEWERS' COMMENTS

Reviewer #1 (Remarks to the Author):

The authors have addressed all the concerns and issues raised in the initial review.

- Ross Hardison

Reviewer #2 (Remarks to the Author):

I thank the Authors for having addressed all the points.

Response to the Reviewer comments

REVIEWERS' COMMENTS

Reviewer #1 (Remarks to the Author):

The authors have addressed all the concerns and issues raised in the initial review.

- Ross Hardison

Reviewer #2 (Remarks to the Author):

I thank the Authors for having addressed all the points.

We thank the Reviewers for their supportive comments on our work.